# Gaze-to-text Generation: Beyond Categorical Decoding of Human Attention

## Abstract

We introduce a novel learning problem: decoding gaze into natural language descriptions of human goals across diverse visual tasks. Unlike prior work, which frames gaze decoding as a classification task over predefined categories, we formulate it as a generative learning problem: training a model to produce free-form descriptions that capture the rich context, nuance, and open-ended nature of human intentions beyond fixed labels.

To this end, we introduce *Gazette*, the first gaze-to-text decoding framework. Based on multimodal large language models (MLLMs), Gazette learns to decode gaze scanpaths into natural language for goals that may extend beyond categorical labels and require articulation in natural language. To help Gazette filter out individual differences in gaze behavior and learn the goal-specific spatiotemporal dynamics crucial for generating accurate natural language goal descriptions, we propose a novel strategy that leverages the encyclopedic knowledge and reasoning abilities of a large language model to synthesize natural language explanations of goal-directed attentional behavior called *think-aloud transcripts*. Instruction tuning on these synthetic narratives allows Gazette to achieve state-of-the-art performance in gaze decoding across multiple tasks, demonstrating its generalizability and versatility, thereby enabling gaze to serve as a powerful, non-intrusive cue for inferring human goals and intentions in diverse scenarios.

## 1 Introduction

Gaze decoding offers a non-intrusive and practical means of inferring human attention and intent, as a valuable alternative to neural decoding from EEG (Hollenstein et al., 2021; Daly, 2023), fMRI (Xia et al., 2024; Takagi & Nishimoto, 2023), MEG (Défossez et al., 2023; Benchetrit et al., 2023), and ECoG (Chestek et al., 2013; Komeiji et al., 2024) that often need bulky, expensive, and invasive equipment. As eye-tracking technology becomes increasingly accessible and accurate, across both wearable and non-wearable devices, the utility of gaze decoding grows, enabling a wide range of applications including user engagement analysis (Bühler et al., 2024; Khokhar et al., 2019), driver monitoring (Tawari et al., 2014), human-robot interaction (Li & Zhang, 2017), hands-free assistive technologies (Perfect et al., 2020), and psychological diagnosis (Liaqat et al., 2021). However, existing gaze decoding approaches (Barz et al., 2020; Sattar et al., 2015; 2017; 2020; Nishiyasu & Sato, 2024; Wang et al., 2024a), cannot easily decode richer, more expressive descriptions of human intent expressed by natural language, as they focus on goal-directed visual tasks based on predefined labels, notably categorical visual search. Such approaches are often inadequate in real-world scenarios where capturing broader linguistic context is essential. For example, analyzing user engagement on an e-commerce website may benefit from natural language descriptions of user gaze patterns, as many ads or products might be novel, and cannot be captured by a fixed label set required by categorical approaches. Furthermore, using natural language to decode user gaze instead of using categorical labels like "chair" might reveal user engagement with products described as "modern minimalist chairs with wooden legs", informing the downstream recommendation system of crucial, fine-grained nuances, enabling them to suggest visually and stylistically similar items even if they are newly listed and not in any predefined category—potentially improving user experience.

To bridge this gap, we study the task of decoding a person's attention to infer their goal, where the goal ranges from a finite set of categories represented by simple labels (*e.g.,* "bowl", "car") to

free-flowing text comprising unstructured sequences of natural language, rich in context and nuance (*e.g.,* "red bowl to the left of the cup in the middle"). The input for our task is an image $\mathbf{I}$ and a language instruction $\mathbf{T}_{GazeDec}$ textually representing the gaze scanpath $\mathbf{S}$, and the output is text $\mathbf{D}$ describing the *cognitive context* of the human. We define cognitive context at both coarse and fine levels. The coarser level is *behavior type*, such as whether a person is engaged in a visual search, object referral, or visual question answering (VQA) task. The finer level consists of a person's specific goal, such as the category label of a search target, and free-flowing unstructured text in the form a referring expression, or a question. To tackle this decoding task, we propose the ***Gaze-to-text*** generation model or *Gazette*, a multimodal large language model (MLLM) based text-generative framework, that decodes goal-directed human attention across a wide range of gaze behaviors.

Building on the broad knowledge encoded in foundational MLLMs (Liu et al., 2023; Li et al., 2024c) and the demonstrated effectiveness of instruction tuning to adapt them to downstream tasks, it is natural to extend this strategy to train a gaze-to-text decoding model. However, this task presents unique challenges, as decoding goals from gaze is inherently difficult – gaze behavior is shaped not only by task-driven goals but also by individual differences among viewers. Since foundational MLLMs lack domain-specific knowledge in specialized areas (Hamza et al., 2025; Duan et al., 2024; Mohbat & Zaki, 2024) like gaze behavior, they struggle to disentangle goal-relevant signals from these variations, yielding poor performance when trained solely on the primary gaze decoding task.

To address this limitation and improve model reasoning, we propose a novel approach that encourages the model to focus specifically on attentional goals while abstracting away individual differences. We construct auxiliary instruction tuning data that trains Gazette to explicitly learn *top-down attention allocation strategies* by generating *think-aloud transcripts* (Ericsson & Simon, 1984; van Someren et al., 1994) – narratives that highlight attentional patterns and information relevant to the top-down attentional goal while filtering out individual variability. To annotate these think-aloud transcripts, we leverage the encyclopedic knowledge and reasoning abilities of GPT-4 (Achiam et al., 2023), a multi-billion-parameter large language model. We introduce a novel prompting strategy grounded in our hypothesis that, for a given visual stimulus (e.g., an image), the attentional goal shared across participants can be inferred from the commonalities in their otherwise diverse gaze patterns.

We show empirically that learning to generate think-aloud transcripts annotated by the teacher model, *i.e.*, GPT-4, enhances decoding performance across multiple gaze behaviors. By tuning Gazette to deeply understand the attention allocation strategies, we encourage it to focus on the goal-specific portions of the scanpath, effectively guiding the primary gaze decoding task. In summary:

1. We introduce the task of unconstrained decoding of goal-directed attention, where top-down goals are expressed in natural language, supporting a wide range of human gaze behaviors.

2. We propose *Gazette*, a novel text-generative MLLM-based framework, that is instruction tuned to decode a scanpath of gaze fixations (during image viewing) to natural language.

3. We further enhance Gazette by instruction tuning on an auxiliary *think-aloud transcript* generation task for identifying goal-specific attentional patterns within scanpaths.

4. We derive ground truth for think-aloud transcripts by prompting GPT-4 using a novel prompting strategy that exploits commonalities in gaze behavior of multiple observers engaged in the same top-down attentional task and visual stimulus.

## 2 RELATED WORK

**Goal-directed Human Attention.** Goal-directed attention, in contrast to bottom-up attention (Itti et al., 1998; Masciocchi et al., 2009), is the top-down control exerted by frontal-parietal brain areas that modulates processing of sensory input based on current task demand, prior knowledge and expectations (Henderson et al., 2007; Koehler et al., 2014). Several datasets have been collected to study various facets of goal-directed gaze behavior. COCO-Search18 (Chen et al., 2021) is a dataset of visual search gaze fixations from 10 participants performing Target-Present/Absent tasks on 6,202 natural images spanning 18 target categories. RefCOCO-Gaze (Mondal et al., 2024) contains gaze scanpaths from 220 participants viewing 2,094 COCO images while simultaneously hearing corresponding referring expressions grounding objects in the images. AiR-D (Chen et al., 2020) is a dataset with scanpaths of 20 participants performing the visual question answering (VQA) task for 195 image–question pairs. In this study, we use COCO-Search18, RefCOCO-Gaze and AiR-D.

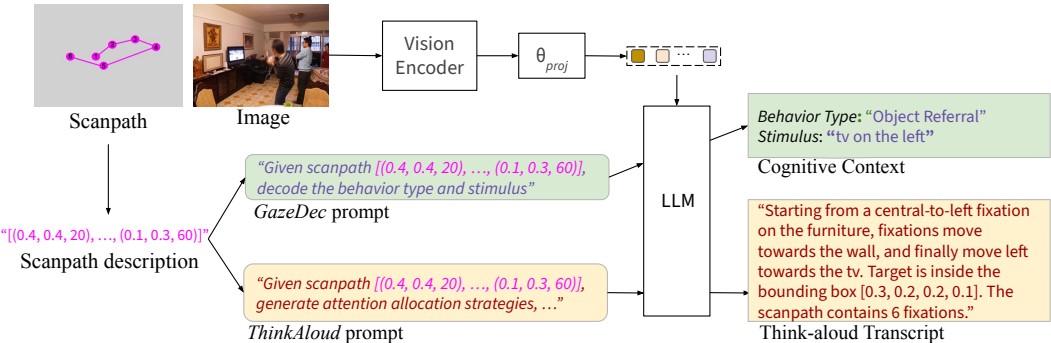

Figure 1: **Gazette**: A text-generative decoding framework for top-down attention. For an input of an image and a language instruction conveying scanpath information, a textual response is generated by the Multimodal LLM comprised of a Vision Encoder, an LLM, and a linear projection $\theta_{proj}$ interfacing the Vision Encoder and the LLM. Language instruction can correspond to either the primary gaze decoding task *GazeDec*, or the auxiliary think-aloud transcript generation task *ThinkAloud*.

**Gaze Decoding.** Gaze measured by eye-trackers can be a noninvasive means of understanding human intention. Yarbus (Yarbus, 1967b) showed that eye movements depend strongly on the viewer's task, with fixations varying across instructions even for the same visual stimulus, *e.g.,* image. Prior work has explored decoding goal-directed tasks from eye movements (Zelinsky et al., 2008; 2013; Bahle et al., 2017; Borji & Itti, 2014; Borji et al., 2015), including reconstructing images from fixations (Wang et al., 2019; Strohm et al., 2021; 2023a;b) and inferring user tasks or activities from gaze (Bulling et al., 2010; Bektaş et al., 2024; Hu et al., 2021b; Steil & Bulling, 2015; Chen et al., 2022). Sattar et al. (Sattar et al., 2015; 2017; 2020) studied search target detection for both images and categories, and later methods predicted search targets from gaze using pre-trained CNN features (Stauden et al., 2018; Barz et al., 2020). The Gaze Scanpath Transformer (GST) (Nishiyasu & Sato, 2024) model incorporates category semantics into scanpath information to predict visual search targets using the COCO-Search18 dataset. GazeGNN (Wang et al., 2024a) is a radiological image diagnosis model, that can be adapted for gaze target decoding. These methods frame decoding as classification – selecting a single mental state or goal from alternatives – which limits their real-world applicability. In a related work, Chen *et al.* (Chen et al., 2024) jointly predicted gaze scanpaths and generated natural-language explanations for each fixation (ignoring sequential dynamics of scanpaths). However, our work accepts human scanpaths as an input, not output, and learns to generate the cognitive context attached to top-down attention. To our knowledge, our proposed framework, Gazette, is the first gaze-to-text decoding method.

**Instruction Tuning of MLLMs.** Instruction tuning or supervised fine-tuning (SFT) for multimodal large language models (MLLMs) adapts pre-trained vision-language architectures for downstream tasks by training on instruction-response pairs, enhancing model performance on specific tasks and general instruction-following (Liu et al., 2023; 2024; Ranasinghe et al., 2024). Methods like LLaVA (Liu et al., 2023) and VIGC (Wang et al., 2024b) generate large-scale visual instruction datasets, often with proprietary LLMs like GPT-4 endowed with image context expressed via object bounding box annotations in the scene. Visual instruction tuning has been applied in diverse domains. Instruction-tuned MLLMs have excelled in robotics (Driess et al., 2023; Li et al., 2024c), healthcare (Singhal et al., 2023), education (Li et al., 2024a), and e-commerce (Liu et al., 2024), demonstrating how instruction tuning transforms general-purpose MLLMs into specialized, domain-adapted agents. Inspired by these works, we curate the first instruction tuning dataset for gaze, enabling the adaptation of general-purpose MLLMs for the decoding of gaze behavior.

## 3 *Gazette*: GAZE-TO-TEXT DECODING OF HUMAN ATTENTION

Existing gaze decoding methods for visual search tasks (Barz et al., 2020; Nishiyasu & Sato, 2024) adopt an $K$-way classification framework to select one of the $K$ predefined search target categories. However, this severely restricts the applicability and extensibility of these methods to a wider range of gaze behaviors, particularly those that have complex attribute and contextual information about

objects, *e.g.,* object referral (Mondal et al., 2024) and VQA (Chen et al., 2020), needing natural language specification. To overcome this limitation, we adopt an MLLM-based text-generative framework for gaze decoding, the *Gaze-to-text* generation model or *Gazette* (Fig. 1). Gazette extends the LLaVA-1.5-7B (Liu et al., 2024) model and is fine-tuned on our novel visual instruction tuning data consisting of tasks tightly linked to gaze understanding. The original LLaVA (Liu et al., 2023) model integrates a frozen pre-trained *Vision Encoder* (Radford et al., 2021) with a large language model (*LLM*) via a lightweight MLP projection $\theta_{proj}$ that projects visual features in the LLM's token embedding space. This "visual prefix" and a language instruction prompt from the user are concatenated and processed by the LLM which then generates language tokens autoregressively to form the textual response. Following (Ranasinghe et al., 2024; Liu et al., 2023; Li et al., 2024c) that encoded scene objects and sensorimotor signals using raw text, we represent the scanpath textually and embed it within the language instruction. In the context of the image, the primary task of Gazette is to textually decode the cognitive context represented in a hierarchical structure containing two facets underlying top-down attentional control: (i) the coarser facet *behavior type*, *i.e.,* the *type* of goal-directed behavior (Target-Present Visual Search, Target-Absent Visual Search, object referral, VQA), and (ii) the finer facet *stimulus* outlining the specifics of the top-down goal. The stimuli for visual search is the target category (*e.g.,* "car", "bottle"), for object referral is the referring expression (*e.g.,* "red car on the right"), and for VQA is the question (*e.g.,* "what vegetables are on the counter?").

Although predicting behavior type (*i.e.,* the coarse-level specification of gaze behavior) can be trivial owing to artifacts from different behavioral data collection setups, decoding the fine-grained stimulus is a hard task. Naively fine-tuning MLLMs on the primary task alone can be suboptimal since general-purpose MLLMs seem to lack prior understanding of human attention control and are unable to disentangle goal-specific information from individual differences within human scanpaths. Hence, we constructed auxiliary instruction tuning data which involves learning *attention allocation strategies* associated with goal-directed behavior by generating *think-aloud transcripts* detailing the goal-specific information and attentional patterns. We derive ground truth for think-aloud transcripts by prompting GPT-4 with a novel strategy that uses scanpaths of multiple participants sharing a common cognitive context to derive a natural language explanation of the latent attentional processes.

### 3.1 GAZE SCANPATH DECODING (*GazeDec*)

We address the extensibility-related shortcomings of existing classification-based gaze decoding frameworks by formulating the gaze decoding task as an image+text-to-text problem, where the input is a combination of an image $\mathbf{I}$ and text $\mathbf{T}_{GazeDec}$ detailing the scanpath $\mathbf{S}$, and the decoded output is another text $\mathbf{D}$ describing the cognitive context. We call this primary task of Gazette *GazeDec*. For an image $\mathbf{I} \in \mathbb{R}^{3 \times H \times W}$ of dimensions $H$ and $W$, a gaze scanpath $\mathbf{S} = \{f_i | i = 0, \ldots, N-1\}$ is a sequence of $N$ eye fixations $f_i = (x_i, y_i, t_i)$, where $f_i$ is represented by three parameters: horizontal location $x_i$, vertical location $y_i$, and fixation duration $t_i$. We normalize $x_i$ and $y_i$ to the range of $[0, 1]$ to ensure scale invariance across datasets. As in previous visual instruction tuning work (Li et al., 2024c; Liu et al., 2024; Ranasinghe et al., 2024) that represented bounding boxes and sensorimotor signals in raw text, we represent $\mathbf{S}$ textually using a prompt template to produce a language instruction prompt $\mathbf{T}_{GazeDec}$. Both $\mathbf{I}$ and $\mathbf{T}_{GazeDec}$ constitute the input to Gazette, which processes the input and generates a textual response $\mathbf{D}$ describing the cognitive context. $\mathbf{D}$ can be disentangled into $\mathbf{D}_{type}$ and $\mathbf{D}_{goal}$ containing the behavior type and stimulus, respectively, via parsing.

### 3.2 THINK-ALOUD TRANSCRIPT GENERATION (*ThinkAloud*)

To endow Gazette with in-depth knowledge of top-down attention processes, we devise a think-aloud transcript generation task, or *ThinkAloud* in short. In think-aloud protocol analysis (Fonteyn et al., 1993), transcripts are composed of "idea units", the smallest semantically coherent chunks of a verbal report, each representing a single thought or proposition. Similarly, our think-aloud transcript has the following idea units: (i) top-down attention allocation explanation, (ii) target location, and (iii) scanpath length. Similar to *GazeDec*, the input is image $\mathbf{I}$ and language instruction prompt $\mathbf{T}_{ThinkAloud}$ embodying the scanpath $\mathbf{S}$, and the decoded output is the think-aloud transcript $\mathbf{TaT}$.

As existing gaze datasets do not contain attention allocation strategies for top-down goal-specific fixation patterns, we must generate these strategies synthetically. Observers' eye movement patterns when viewing the same scene vary dramatically depending on the specific task (Yarbus, 1967a),

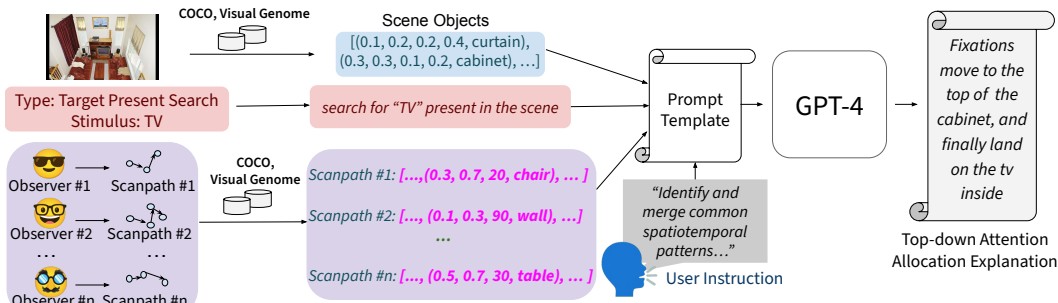

Figure 2: A novel prompting strategy for GPT-4 to extract top-down attention allocation explanation for think-aloud transcript generation task (*ThinkAloud*). This strategy instructs GPT-4 to summarize common spatiotemporal patterns in scanpaths of $n$ participants, given the task type and goal, along with scene information via scene object bounding boxes from COCO (Lin et al., 2014) and Visual Genome (Krishna et al., 2017). The response is used to construct the think-aloud transcript.

suggesting that goal-specific information is embodied by eye movement patterns. However, human visual attention is driven by many factors such as bottom-up salience, top-down goals, and idiosyncracies (or "attentional fingerprints," *e.g.,* genetic factors, personality traits, neurodiversity), making it non-trivial to identify goal-specific information. We thus hypothesize that scanpaths of *different* people sharing a common top-down goal share common patterns,indicative of the top-down goal. Based on this hypothesis, we devise a novel prompting strategy to query GPT-4 (Achiam et al., 2023) to derive pseudo-annotations for think-aloud transcripts corresponding to our gaze data.

**Novel prompting strategy for GPT-4 to annotate top-down attention allocation explanation.** In addition to the attentional goal, human top-down attention control is affected by several individual factors such as age, ocular health, and neurodiversity. These individual factors are analogous to "style" while the goal is analogous to "content". We hypothesize that this goal-specific information is contained within the *common spatiotemporal patterns* within scanpaths of multiple participants engaged in the same attentional task for the same visual stimulus, *e.g.,* image. These common patterns are likely due to the common goal shared by the participants, potentially unaffected by individual-level idiosyncrasies and noise. Since LLMs excel at identifying patterns within data (Mirchandani et al., 2023; Weber, 2024b) and have encyclopedic world knowledge, we use GPT-4 (Achiam et al., 2023) to analyze scanpaths from distinct participants sharing the same cognitive context for the same image, and summarize the common spatiotemporal patterns (see Fig. 2). As noted in (Wang et al., 2024b), currently available accessible MLLMs are less capable and more expensive than LLMs. Hence, as in LLaVA (Liu et al., 2023), we use text-only GPT-4 for its accessibility, cost-effectiveness, and efficiency. We provide scene context in the form of bounding-box annotations of objects in the scene derived from COCO (Lin et al., 2014) for COCO-Search18, RefCOCO-Gaze, and COCO-originated images in the AiR-D dataset, and from Visual Genome (Krishna et al., 2017) scene graphs for Flickr-originated images in AiR-D dataset sourced from GQA dataset (Hudson & Manning, 2019). Again, to counter scale variation, we normalize bounding-box parameters. Additionally, we provide scanpath information for all observers in the form of normalized fixation co-ordinates, raw fixation durations and category labels of the fixated objects (derived from COCO and Visual Genome bounding box annotations). Finally, cognitive context is also included in the prompt. Prompt details are given in Appendix C. GPT-4's response reflects common patterns across scanpaths of several participants, which we hypothesize indicates the top-down attentional allocation strategies utilized by the human visual system, therefore serving as pseudo-annotation for the "top-down attention allocation explanation" idea unit of the think-aloud transcript. Because this response captures the *common* patterns, it remains consistent across *all* scanpaths from multiple participants, and can be used as psuedo-annotation for scanpaths from *each* individual participant. Additionally, following previous visual instruction tuning research (Ranasinghe et al., 2024; Li et al., 2024c) adapting MLLMs for specialized tasks, we propose two fundamental scanpath comprehension tasks as "idea units" in the think-aloud transcript: target localization, and counting fixations in a scanpath. Since MLLMs struggle in object localization (Ranasinghe et al., 2024), a crucial task underlying gaze behavior for visual search and object referral (Yang et al., 2022; Mondal et al., 2023), we posit that Gazette will

benefit from learning target localization, particularly for Visual Search and Object Referral tasks. Following previous MLLM literature (Ranasinghe et al., 2024; Liu et al., 2024; Li et al., 2024c), we normalize bounding box parameters $[x, y, w, h]$ (where $x, y$ locate the upper-left corner, and $w$ and $h$ are the width and height of the bounding box, respectively) to deal with scale variation. This task also accounts for target-absent cases (null scenario), where Gazette must predict the absence of the target instead of a bounding box. The second task is inspired by previous research (Tsang et al., 2010; Williams & Castelhano, 2019; Sabab et al., 2022) suggesting that scanpath length indicates many properties of gaze behavior, *e.g.,* the degree of exploratory compared to focused behavior. Prompts for both *GazeDec* and *ThinkAloud* along with human evaluation of the *ThinkAloud* is in Appendix C.

### 3.3 MODEL TRAINING AND INFERENCE

The following are training and inference procedures for Gazette. More details are in Appendix D.

**Training.** We initialize our model with pre-trained LLaVA-1.5-7B model (Liu et al., 2024) weights, and fine-tune it via Low-Rank Adaptation (LoRA) (Hu et al., 2021a).This reduces the number of trainable parameters, mitigating overfitting on limited gaze training data. We use an auto-regressive language modeling objective (Liu et al., 2023) for instruction tuning. Both sets of instructions, *i.e.,* $\mathbf{T}_{GazeDec}$ for *GazeDec*, and $\mathbf{T}_{ThinkAloud}$ for *ThinkAloud*, are used for instruction tuning.

**Inference.** We prompt Gazette using $\mathbf{T}_{GazeDec}$ and decode the response text using greedy decoding strategy. The response text is then parsed to obtain the gaze behavior type $\mathbf{D}_{type}$ and the stimulus $\mathbf{D}_{goal}$. $\mathbf{D}_{type}$ is encoded using a language encoder (Wang et al., 2020) and matched to label vocabulary by computing cosine similarities between each text embedding and each label's language embedding, where label vocabulary is {"Target-Present Search", "Target-Absent Search", "Object Referral", or "Visual Question Answering"}. Similarly, for COCO-Search18, $\mathbf{D}_{goal}$ is matched with the 18 target categories of the dataset.

## 4 EXPERIMENTS

*Gazette* can decode gaze scanpaths for multiple gaze behaviors, including categorical visual search, object referral, and visual question answering (VQA). In this section, we evaluate Gazette's gaze decoding capabilities using a diverse array of metrics across multiple gaze behaviors. Gaze behavior type prediction is trivial for Gazette (see Suppl.), perhaps because it recognizes artifacts in scanpaths from distinct behavioral data collection setups. Here, we focus on stimulus decoding.

We train our model on two NVIDIA RTX A6000 cores with a total batch size of 32 and a learning rate of 2e-5. Training is done on the training splits of COCO-Search18 (both Target-Present and Target-Absent trials), RefCOCO-Gaze and AiR-D, and evaluation is done on the corresponding test splits. To evaluate and analyze the efficacy of our proposed auxiliary task, we compare performances of our full model Gazette trained on both sets of instructions $\mathbf{T}_{GazeDec}$ and $\mathbf{T}_{ThinkAloud}$ (corresponding to *GazeDec* and *ThinkAloud*, respectively), and a variant where the MLLM is trained only on $\mathbf{T}_{GazeDec}$ but not $\mathbf{T}_{ThinkAloud}$. The latter variant is simply denoted as "*w/o ThinkAloud*".

While classification-based gaze decoding methods exist (Wang et al., 2024a; Nishiyasu & Sato, 2024), Gazette is the first gaze-to-text model to our knowledge, so we constructed baselines based on SOTA LLaVA (Liu et al., 2024)model. For Object Referral and Target-Present Visual Search, a majority of last fixations land on the target, so we fine-tune LLaVA-1.5 to describe the object where the final fixation lands, to yield the gaze stimulus. We call this baseline *LLaVA-last*. Another baseline is a frozen *LLaVA-1.5* (Liu et al., 2024) model prompted with scanpath information and instructions to describe the goal. We also compare Gazette with baselines GST (Nishiyasu & Sato, 2024) and GazeGNN (Wang et al., 2024a) (adapted by us for visual search) in visual search decoding with COCO-Search18 (Chen et al., 2021). More details of Gazette and baselines are in the Appendix D.

### 4.1 EVALUATION METRICS

We curated a broad set of metrics suited to gaze behavior. For visual search, gaze is decoded by selecting a target category from a predefined set, while for object referral and VQA it is decoded into free-form referring expressions and questions. Accordingly, the metrics for object referral and VQA differ from those for visual search, as detailed below.

### 4.1.1 TEXT GENERATION EVALUATION METRICS FOR OBJECT REFERRAL AND VQA

We evaluate model text-generative capabilities on Object Referral gaze decoding and VQA gaze decoding using two paradigms: (1) using standard lexical overlap metrics, and (2) using GPT-4 to assess generated texts using a set of abstract rubrics, also known as LLM-as-a-Judge paradigm.

**Lexical Overlap Metrics.** To assess the text generation-based decoding capabilities of Gazette, we use a broad set of standard metrics used to evaluate the lexical overlap of generated text from text generation models with the ground truth text. **BLEU-1 to BLEU-4** (Papineni et al., 2002) evaluate machine-generated text by comparing its n-gram precision (n=1–4) against reference translations and applies a brevity penalty to discourage overly short outputs. **METEOR** (Banerjee & Lavie, 2005) aligns candidate and reference translations via exact, stem, synonym, and paraphrase matches, then computes a recall-weighted harmonic mean of unigram precision and recall with a fragmentation penalty to preserve word order. **ROUGE-L** (Lin, 2004) measures the longest common subsequence between candidate and reference texts, combining recall and precision into an F-score to capture sentence-level structure and word-order overlap. **CIDEr** (Vedantam et al., 2015) captures human consensus by computing the TF–IDF–weighted cosine similarity of n-gram vectors between a candidate and multiple references, highlighting terms frequent in the candidate and distinctive in the references. While there is only one reference question for VQA samples in AiR-D, there are multiple reference referring expressions in RefCOCO (Yu et al., 2016) annotated by multiple annotators for the same object in an image. This allows us to compute lexical overlap metrics for each referring expression as the candidate, while reserving the others as ground truth. This is repeated for all referring expressions; the average is the RefCOCO Inter-Annotator Consistency (**RefCOCO-IAC**). The consistency values approximate the variability among humans annotating the same object, serving as a noise ceiling (similar to IOC values in behavioral literature).

**LLM-as-a-Judge.** In this evaluation paradigm popularized by recent text-generative research (Liu et al., 2023; Zheng et al., 2023), a very large language model (such as GPT-4) is used as an external evaluator to assess the quality of the generated outputs, providing a scalable and cost-effective alternative to human evaluation. Additionally, this allows more interpretable evaluation rubrics to be used (*e.g.,* fluency, helpfulness, relevance), allowing for a more nuanced, and context-aware, approach to assessing model-generated text that might not be possible with standard lexical overlap metrics. For each scanpath, GPT-4 is prompted to evaluate texts using a scale from 1 to 10 for each rubric, where 1 is poor and 10 is excellent. The prompt also contains these key information: **(1)** referring expressions/questions generated by all models to be evaluated, **(2)** ground truth referring expressions/questions, **(3)** scene context in the form of bounding boxes of every object in the scene (similar to the method described in Sec. 3.2), **(4)** rubrics to evaluate the generated texts on. The rubrics are distinct for object referral and VQA and are detailed below and in Appendix C).

For object referral, we use the following rubrics. **Expression Overlap** - How much does the generated referring expression match the ground-truth referring expressions, especially in terms of coverage of the entities, their attributes and spatial relationships? **Referential Equivalence** - Does either of the ground-truth expressions refer to the same object in the image as the generated expression? **Category Correctness** - Is the type or category of the referred object mentioned correctly?

For VQA, we use the following rubrics. **Question Overlap** - How much does the generated question match the ground truth question, especially in terms of coverage of the entities, their attributes and spatial relationships? **Answer Equivalence** - Does the generated question and the ground-truth question have the same correct answer on the image?

### 4.1.2 TARGET CATEGORY PREDICTION METRICS FOR VISUAL SEARCH TASKS

Since there are scanpaths for 18 distinct target categories in COCO-Search18, we evaluate decoding methods according to their ability to correctly distinguish these target categories, as quantified by **precision**, **recall**, $F_1$ **score**, and **accuracy**. Since the COCO-Search18 test set is imbalanced in terms of number of scanpaths per category, we average the reported precision, recall and $F_1$ scores across the 18 COCO-Search18 search target categories, equally weighting each category. On the other hand, accuracy does not account for this imbalance, but we include it to compare with accuracy value reported for SOTA method GST (Nishiyasu & Sato, 2024) whose implementation is not public.

## 4.2 RESULTS

Our evaluation assesses the efficacy of Gazette in decoding gaze scanpaths originating from four top-down attention behavior tasks: Task A – Object Referral, Task B – Visual Question Answering (VQA), Task C – Target-Present Visual Search, and Task D – Target-Absent Visual Search. First, we evaluate generative gaze decoding on Object Referral (using RefCOCO-Gaze (Mondal et al., 2023) dataset) and VQA (using Air-D (Chen et al., 2020) dataset) tasks under two paradigms: (1) Lexical Overlap Metric-based evaluation (Table 1) and (2) LLM-as-a-Judge-based evaluation (Table 2).

Table 1: Performance of Gazette and baselines on Task A – Object Referral Gaze Decoding evaluated using RefCOCO-Gaze (Mondal et al., 2024), and Task B – VQA Gaze Decoding evaluated using AiR-D (Chen et al., 2020) on the basis of lexical overlap metrics. Best results are highlighted in bold. Results exceeding RefCOCO Inter-Annotator Consistency (RefCOCO-IAC) values are underlined. Percentage improvements of Gazette over variant w/o *ThinkAloud* are provided in parentheses.

| Task | Method | BLEU-1 | BLEU-2 | BLEU-3 | BLEU-4 | METEOR | ROUGE-L | CiDeR |
|------|--------|--------|--------|--------|--------|--------|---------|-------|
| | RefCOCO-IAC | 0.500 | 0.282 | 0.148 | 0.057 | 0.244 | 0.452 | 1.115 |
| Object Referral (Task A) | LLaVA-1.5 (Liu et al., 2024) | 0.066 | 0.018 | 0.006 | 0.0 | 0.077 | 0.105 | 0.062 |
| | *LLaVA-last* | 0.170 | 0.080 | 0.039 | 0.014 | 0.113 | 0.221 | 0.113 |
| | *w/o ThinkAloud* | 0.479 | 0.263 | 0.145 | 0.070 | 0.232 | 0.443 | 0.872 |
| | *Gazette* | **0.519** | **0.305** | **0.175** | **0.098** | **0.248** | **0.480** | **0.974** |
| | | (+8.36%) | (+15.97%) | (+20.69%) | (+40.00%) | (+6.90%) | (+8.36%) | (+11.70%) |
| VQA (Task B) | LLaVA-1.5 (Liu et al., 2024) | 0.124 | 0.030 | 0.012 | 0.006 | 0.039 | 0.103 | 0.047 |
| | *w/o ThinkAloud* | 0.329 | 0.222 | 0.144 | 0.095 | 0.147 | 0.286 | 0.263 |
| | *Gazette* | **0.364** | **0.268** | **0.202** | **0.159** | **0.160** | **0.324** | **0.367** |
| | | (+10.64%) | (+20.72%) | (+40.28%) | (+67.37%) | (+8.84%) | (+13.29%) | (+39.54%) |

Table 2: Performance of Gazette and baselines on Task A – Object Referral Gaze Decoding evaluated using RefCOCO-Gaze dataset (Mondal et al., 2024), and Task B – VQA Gaze Decoding evaluated using AiR-D dataset (Chen et al., 2020) by GPT-4 (Achiam et al., 2023) on a scale of 1-10 under the LLM-as-a-Judge setting. The scores are averaged and provided below with best scores in bold.

| | | Method | | |
|------|-------|---------------|-----------------|---------|
| Task | Rubric | *LLaVA-last* | *w/o ThinkAloud* | *Gazette* |
| Object Referral (Task A) | Expression Overlap | 3.515 | 5.019 | **5.980** |
| | Referential Equivalence | 4.100 | 5.771 | **6.638** |
| | Category Correctness | 7.577 | 8.376 | **8.743** |
| VQA (Task B) | Question Overlap | - | 2.103 | **2.792** |
| | Answer Equivalence | - | 1.642 | **2.135** |

As shown in Table 1, Gazette trained with auxiliary *ThinkAloud* task significantly outperforms the variant not trained on *ThinkAloud* ("*w/o ThinkAloud*"), and baselines LLaVA-1.5 (Liu et al., 2024) and *LLaVA-last* for both Task A and Task B. Under the LLM-as-a-Judge paradigm (Table 2), the reported scores follow the same trend seen in lexical overlap metrics, with Gazette outperforming the variant not trained on *ThinkAloud* ("*w/o ThinkAloud*") by large margins on every human-interpretable rubric. This shows the efficacy of deeper reasoning about the top-down attentional processes via the *ThinkAloud* task. *LLaVA-last* baseline performs poorly in Tables 1 and 2, suggesting that complete scanpath provides crucial contextfor referring expression generation. In Appendix A, we explore the individual effects of idea units within think-aloud transcripts, and show that for Object Referral, learning both attention allocation explanation generation and target localization enhances performance. For VQA (where target localization is not relevant), only attention allocation explanation generation is crucial, as it helps Gazette learn the complex reasoning processes underlying VQA.

Next, we focus on evaluating the target prediction capabilities of Gazette (trained with and without *ThinkAloud*), and compare with the baselines GazeGNN (Wang et al., 2024a) (which we adapt for

Table 3: Performance of Gazette and baselines on Task C – Target-Present Visual Search Gaze Decoding, and Task D – Target-Absent Visual Search Gaze Decoding, both evaluated using COCO-Search18 dataset (Chen et al., 2021). Best performance results are highlighted in bold.

| Task | Method | Precision | Recall | $F_1$ | Accuracy |
|---|---|---|---|---|---|
| Target-Present Visual Search (Task C) | LLaVA-1.5 (Liu et al., 2024) | 0.448 | 0.436 | 0.406 | 0.402 |
| | *LLaVA-last* | 0.688 | 0.649 | 0.628 | 0.618 |
| | GazeGNN (Wang et al., 2024a) | 0.338 | 0.345 | 0.319 | 0.335 |
| | GST (Nishiyasu & Sato, 2024) | - | - | - | 0.544 |
| | *w/o ThinkAloud* | 0.768 | 0.746 | 0.742 | 0.742 |
| | *Gazette* | **0.786** | **0.775** | **0.775** | **0.773** |
| Target-Absent Visual Search (Task D) | LLaVA-1.5 (Liu et al., 2024) | 0.128 | 0.102 | 0.077 | 0.088 |
| | GazeGNN (Wang et al., 2024a) | 0.241 | 0.251 | 0.218 | 0.270 |
| | GST (Nishiyasu & Sato, 2024) | - | - | - | 0.385 |
| | *w/o ThinkAloud* | 0.437 | 0.424 | 0.420 | **0.445** |
| | *Gazette* | **0.438** | **0.426** | **0.424** | 0.443 |

target prediction), GST (Nishiyasu & Sato, 2024), LLaVA-1.5 (Liu et al., 2024), and *LLaVA-last*, on the COCO-Search18 benchmark (Chen et al., 2021). The results are shown in Table 3. The implementation for GST is not publicly available, therefore we are unable to furnish metrics not reported by the authors (*i.e.,* precision, recall, $F_1$). Gazette significantly outperforms previous methods in all available metrics for both Target-Present and Target-Absent Visual Search tasks. *Gazette-last* fares well in recognizing Target-Present gaze targets, but still lags behind Gazette, suggesting the importance of context provided by scanpaths. In Appendix A, we show that akin to Object Referral, both target localization and attention allocation explanation generation idea units are important for Target-Present search gaze decoding. For Target-Absent trials, training with *ThinkAloud* results in modest performance improvements. We attribute this to poor agreement between observers in Target-Absent search (Yang et al., 2022), where gaze behavior increasingly resembles free-viewing with progression of search (Chen et al., 2022), resulting in low commonality in gaze patterns across participants, potentially affecting GPT-4's responses when prompted by our strategy. In Appendix F, we qualitatively analyze model-decoded cognitive contexts and think-aloud transcripts, shedding light on Gazette's understanding of human attention behavior. We also provide a human evaluation of the GPT-4 generated transcripts in Appendix C.2. This paper's use of LLMs is detailed in Appendix E.

## 5 CONCLUSION

In this work, we explored the capabilities of Multimodal LLMs in understanding human attention behavior, specifically top-down attention control. We proposed *Gazette*, a novel text-generative gaze decoding framework to decode a wide array of top-down attention behaviors. Rather than naively instruction-tuning MLLMs on the primary gaze decoding task, we built an auxiliary dataset for generating top-down attentional allocation explanations – termed *think-aloud transcripts* – to encourage Gazette to explicitly separate goal-specific attentional dynamics from individual idiosyncrasies. To generate pseudo-annotations for these think-aloud transcripts, we prompted GPT-4 using a novel prompting strategy which exploits commonalities in gaze patterns across participants engaged in the same attentional task for the same image. We showed that when trained with a combination of our proposed primary and auxiliary tasks, Gazette achieved significant performance boost in both generative and predictive gaze decoding tasks over naive instruction tuning strategies and SOTA methods, as evaluated by our rigorous evaluation scheme. Although we showed generalization of Gazette to a diverse set of top-down attentional behaviors, it can potentially be extended to applications in psychological analysis, driving, and assistive healthcare, which require decoding human mental states. We expect that Gazette, along with our novel prompting strategy, will inspire future work to explore new avenues of gaze understanding using LLMs and MLLMs.

**Limitations.** Our GPT-4 prompting strategy relies on multi-subject data, limiting its applicability in single-subject settings. Furthermore, its performance may degrade on tasks with high variability in gaze behavior, such as Target-Absent Search, where shared attentional patterns are weak or scarce.

**Reproducibility Statement.** We provide implementation details of our model, Gazette, and our GPT-4 prompting strategy in Sec. 4 and Appendices C and D. Code will be released upon publication.

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

# APPENDIX

## A PROBING IDEA UNITS IN THINK-ALOUD TRANSCRIPTS

In Sec. 3.2, we mentioned that the think-aloud transcript contain three "idea units".In this section, we will discuss their impacts on gaze decoding. Specifically, these three idea units are: (i) top-down attention allocation explanation, which we denote as *GenAttAlloc*, (ii) target location, which we denote as *LocTarget*), and (iii) scanpath length, which we denote as *CountFix*. We probe the effects of these idea units on performance through a series of ablations on the four gaze behaviors in our studies. In each ablation, we either remove or retain each idea unit in the think-aloud transcript. Note that for each ablation, $\mathbf{T}_{ThinkAloud}$ is identical for fair comparison. The results are in Table 4, Table 5, Table 6, and Table 7, for Target-Present Visual Search, Target-Absent Visual Search, Object Referral, and Visual Question Answering (VQA), respectively.

In Table 4, we observe that for Object Referral, while *GenAttAlloc* idea unit (*i.e.,* attention allocation explanation generation sub-task) is crucial, only when it is combined with target localization sub-task, do we achieve best performance. We also note that *CountFix* and *LocTarget* are not sufficient for Gazette to learn how to decode object referral gaze optimally. A similar trend is seen in Table 6 for Target-Present Search, where localization is more important than any other idea unit. We attribute this to short scanpaths for Target-Present Visual Search, where fixations usually land on the target within 1-2 fixations after the initial fixation. On the other hand, *GenAttLoc* embodying the attention allocation explanation generation task is most crucial for VQA Gaze Decoding (see Table 5), with other idea units negatively affecting performance when added to the think-aloud transcript (see row 5 in Table 5). However, for Target-Absent Visual Search (Table 7), predicting the absence of the target (embodied by *LocTarget*) and counting fixations in a scanpath (*CountFix*) seem to improve performance more than *GenAttAlloc* does. We have attributed this to low commonality in gaze patterns across participants for Target-Absent Search, potentially affecting GPT's responses when prompted by our strategy. Overall, we find a combination of all idea units is most beneficial, prompting us to include all of them in the think-aloud transcript annotations.

Table 4: Probing effects of idea units (*GenAttAlloc, LocTarget, CountFix*) in think-aloud transcripts on Object Referral Gaze Decoding. Best results are highlighted in bold. The last row signifies our full model, *Gazette*.

| GenAttAlloc | LocTarget | CountFix | BLEU-1 | BLEU-2 | BLEU-3 | BLEU-4 | METEOR | ROUGE-L | CiDeR |
|:---:|:---:|:---:|:---:|:---:|:---:|:---:|:---:|:---:|:---:|
| × | × | × | 0.479 | 0.263 | 0.145 | 0.070 | 0.232 | 0.443 | 0.872 |
| ✓ | × | ✓ | 0.455 | 0.241 | 0.121 | 0.061 | 0.224 | 0.429 | 0.806 |
| ✓ | ✓ | × | 0.485 | 0.275 | 0.152 | 0.076 | 0.233 | 0.452 | 0.910 |
| × | ✓ | ✓ | 0.494 | 0.272 | 0.153 | 0.085 | 0.239 | 0.446 | 0.918 |
| ✓ | × | × | 0.472 | 0.259 | 0.134 | 0.061 | 0.227 | 0.426 | 0.838 |
| ✓ | ✓ | ✓ | **0.519** | **0.305** | **0.175** | **0.098** | **0.248** | **0.480** | **0.974** |

Table 5: Probing effects of idea units (*GenAttAlloc, LocTarget, CountFix*) in think-aloud transcripts on VQA Gaze Decoding. Best results are highlighted in bold. The last row signifies our full model, *Gazette*.

| GenAttAlloc | LocTarget | CountFix | BLEU-1 | BLEU-2 | BLEU-3 | BLEU-4 | METEOR | ROUGE-L | CiDeR |
|---|---|---|---|---|---|---|---|---|---|
| × | × | × | 0.329 | 0.222 | 0.144 | 0.095 | 0.147 | 0.286 | 0.263 |
| ✓ | × | ✓ | 0.354 | 0.252 | 0.178 | 0.133 | 0.155 | 0.316 | 0.300 |
| ✓ | ✓ | × | 0.364 | 0.264 | 0.193 | 0.147 | 0.163 | 0.330 | 0.310 |
| × | ✓ | ✓ | 0.369 | 0.261 | 0.184 | 0.138 | 0.156 | 0.327 | 0.278 |
| ✓ | × | × | **0.375** | **0.278** | **0.206** | **0.159** | **0.169** | **0.342** | **0.433** |
| ✓ | ✓ | ✓ | 0.364 | 0.268 | 0.202 | **0.159** | 0.160 | 0.324 | 0.367 |

Table 6: Probing effects of idea units (*GenAttAlloc, LocTarget, CountFix*) in think-aloud transcripts on Target-Present Visual Search Gaze Decoding. Best results are highlighted in bold. The last row signifies our full model, *Gazette*.

| GenAttAlloc | LocTarget | CountFix | Precision | Recall | $F_1$ | Accuracy |
|---|---|---|---|---|---|---|
| × | × | × | 0.768 | 0.746 | 0.742 | 0.742 |
| ✓ | × | ✓ | 0.714 | 0.709 | 0.701 | 0.703 |
| ✓ | ✓ | × | 0.759 | 0.753 | 0.749 | 0.748 |
| × | ✓ | ✓ | 0.765 | 0.745 | 0.745 | 0.749 |
| ✓ | × | × | 0.692 | 0.686 | 0.679 | 0.686 |
| ✓ | ✓ | ✓ | **0.786** | **0.775** | **0.775** | **0.773** |

## B  GAZE BEHAVIOR TYPE PREDICTION RESULTS

The gaze behavior type prediction results for Gazette with and without *ThinkAloud* instruction tuning are provided in Table 8. As mentioned in Sec. 4, we find that gaze behavior type prediction is trivial for Gazette, achieving more than 0.90 across precision, recall, and $F_1$ scores, regardless of whether it is trained with or without *ThinkAloud* instructions. We attribute this to the fact that the four gaze behaviors we study are derived from three distinct datasets, collected under different behavioral setups, potentially contributing to artifacts that can easily be recognized by the model without additional supervision from think-aloud transcripts. However, following previous gaze decoding work (Nishiyasu & Sato, 2024) and for the sake of completeness, we retain the coarser behavior type facet of the cognitive context. The main focus of the paper remains decoding the finer stimulus facet of the cognitive context, which has been shown to be a non-trivial task in this paper.

Table 7: Probing effects of idea units (*GenAttAlloc, LocTarget, CountFix*) in think-aloud transcripts on Target-Absent Visual Search Gaze Decoding. Best results are highlighted in bold. The last row signifies our full model, *Gazette*.

| *GenAttAlloc* | *LocTarget* | *CountFix* | **Precision** | **Recall** | **F$_1$** | **Accuracy** |
|---|---|---|---|---|---|---|
| × | × | × | 0.437 | 0.424 | 0.420 | 0.445 |
| ✓ | × | ✓ | 0.421 | 0.409 | 0.409 | 0.427 |
| ✓ | ✓ | × | 0.434 | 0.425 | 0.423 | 0.445 |
| × | ✓ | ✓ | **0.445** | **0.426** | **0.424** | **0.456** |
| ✓ | × | × | 0.422 | 0.418 | 0.412 | 0.438 |
| ✓ | ✓ | ✓ | 0.438 | **0.426** | **0.424** | 0.443 |

Table 8: Performance of Gazette (full model and wø*ThinkAloud* on gaze behavior type prediction as measured by precision, recall, and F$_1$ scores.

| **Task** | **Method** | **Precision** | **Recall** | **F$_1$** |
|---|---|---|---|---|
| Object Referral (Task A) | *w/o ThinkAloud* | 0.99 | 0.95 | 0.97 |
| | *Gazette* | 0.93 | 0.99 | 0.96 |
| Visual Question Answering (Task B) | *w/o ThinkAloud* | 0.99 | 1.00 | 1.00 |
| | *Gazette* | 1.00 | 0.98 | 0.99 |
| Target-Present Visual Search (Task C) | *w/o ThinkAloud* | 0.84 | 0.81 | 0.82 |
| | *Gazette* | 0.85 | 0.83 | 0.84 |
| Target-Absent Visual Search (Task D) | *w/o ThinkAloud* | 0.81 | 0.85 | 0.83 |
| | *Gazette* | 0.83 | 0.84 | 0.84 |
| Tasks A, B, C, and D combined | *w/o ThinkAloud* | 0.91 | 0.90 | 0.90 |
| | *Gazette* | 0.90 | 0.91 | 0.91 |

## C  PROMPTING PROCEDURES FOR GAZETTE AND GPT-4

### C.1  GAZETTE INSTRUCTION TUNING.

The instruction tuning data used to fine-tune Gazette is a combination of two types of instructions for the same scanpath-image input, where the scanpath is generally denoted as a series of fixations (x,y,t): $[(x_0, y_0, t_0), ..., (x_{N-1}, y_{N-1}, t_{N-1})]$. Note that the spatial coordinates x and y are normalized while fixation duration t is unnormalized. An example fixation is: (0.661, 0.112, 272.0).

- **Instruction $\mathbf{T}_{GazeDec}$ for primary task *GazeDec*:**
  We use the following prompt template to construct instruction $\mathbf{T}_{GazeDec}$ for primary task *GazeDec*:

  ```
  Given a scanpath [(x_0,y_0,t_0),    ...,    (x_{N-1},y_{N-1},t_{N-1})], which is a
  list of fixations [x, y, t], with spatial co-ordinates
  x and y normalized between 0 and 1 and t as the fixation
  duration), analyze the sequence in the context of the
  given image to infer the underlying cognitive process
  defined by the top-down task #TASK and the top-down stimulus
  #STIMULUS. Approach the problem through these systematic
  steps:  (1) Identify the task type (Target-Present Search,
  Target-Absent Search, Object Referral, or Visual Question
  Answering) and label it as #TASK. (2) Generate a description
  of the top-down stimulus or goal, marking it as #STIMULUS.
  (3) Format your final response as:  <task> #TASK </task>
  <stimulus> #STIMULUS </stimulus>
  ```

  The ground-truth response is in XML format for easy parsing of the textual response from Gazette to $\mathbf{D}_{type}$ and $\mathbf{D}_{goal}$. A sample response is: `<task> Object Referral </task> <stimulus> red car on the left </stimulus>`

- **Instruction $\mathbf{T}_{ThinkAloud}$ for *ThinkAloud***
  We follow recent instruction tuning work (Li et al., 2024b; Cai et al., 2025) and combine all idea-units of the think-aloud transcript into a single instruction $\mathbf{T}_{ThinkAloud}$, thus avoiding training Gazette on the same image-scanpath pair multiple times that leads to poor generalization on our limited gaze training data. The prompt template for $\mathbf{T}_{ThinkAloud}$ is the following:

  ```
  Given a scanpath [(x_0,y_0,t_0), ..., (x_{N-1},y_{N-1},t_{N-1})], which is a list
  of fixations [x, y, t], with spatial co-ordinates x and y
  normalized between 0 and 1 and t as the fixation duration),
  analyze the sequence in the context of the given image
  to describe the attention allocation strategy #STRATEGY.
  Approach the problem through these systematic steps:  (1)
  Describe the attentive strategy based on fixation density,
  areas of interest, durations, and spatiotemporal patterns
  between fixations and label this text as #STRATEGY. (2)
  Generate #STRATEGY as:  <strategy> #STRATEGY </strategy>"
  ```

### C.2  GPT-4 PROMPTING STRATEGY FOR $\mathbf{T}_{ThinkAloud}$

The prompt template for querying GPT-4 using our prompting strategy is:

*System  prompt*:      `You are a specialist in human visual attention, adept at analyzing images through bounding box annotations and summarizing fixation patterns.`

*User  prompt*:     `Given an image described by objects represented as <category: [x, y, w, h]>:[SCENE_OBJECTS], and 10 humans are searching for a TARGET labeled "bottle"that is present in the`

Stimulus: **Object Referral** for **"car on the right"**.

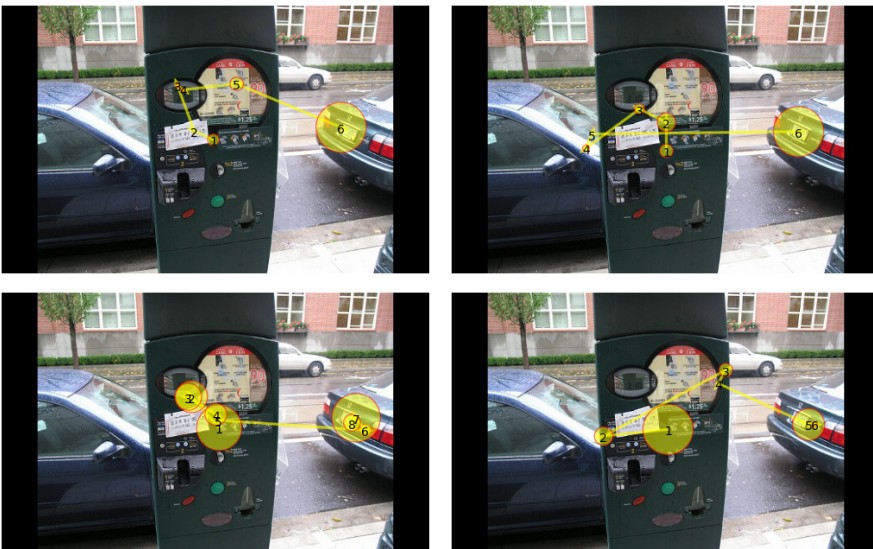

GPT-4 generated: **Most humans initially fixate on the large vertical parking meter spanning a significant portion of the left-central part of the image, then shift focus toward the large car located on the right side, spending varying durations there.**

Figure 3: For the given image and a give top-down task (object referral for referring expression **"car on the right"**, GPT-4 summarizes the common spatiotemporal patterns across several scanpaths (here, we show four scanpaths for the same image-stimulus pair) as per our novel prompting strategy.

```
image.  Eye movements of each human are recorded as a scanpath
- a sequence of fixations, with each fixation represented as
<x-location, y-location , fixation duration, object fixated>:
Human 0 - <fix_0, ..., fix_n>, ..., Human 9 - <fix_0, ..., fix_n>,
identify and concisely merge the most common spatiotemporal
patterns across all scanpaths in one sentence.  Use clear,
unambiguous referring expressions containing attributes and
spatial relationships to refer to objects, and avoid referencing
individual scanpaths, adding extra information, or formatting the
response.
```

Note that [SCENE_OBJECTS] are represented by a list of scene objects, each represented by category of the object and its normalized bounding box, *e.g.,* <bottle:[0.634, 0.068, 0.058, 0.321]>, <chair:[0.519, 0.002, 0.27, 0.304]>. Additionally, each fixation $fix_i$ is represented as normalized x,y co-ordinates, raw fixation duration, and category of the fixated object, *e.g.,* (0.661, 0.112, 272.0, bottle). As mentioned in Sec. 3, the category of the fixated object is derived from COCO and Visual Genome annotations. Through this prompt, we leverage the general pattern-understanding capabilities of GPT-4 as documented by several works (Mirchandani et al., 2023; Weber, 2024a), and do not assume that GPT-4 implicitly understands gaze behavior. Previous work on MLLMs have successfully used GPT-4 for spatial and spatio-temporal synthetic training data generation like LLaVA (Liu et al., 2023) , VIGC (Wang et al., 2024b), Zhang et al. (Zhang et al.). Following their footsteps, we used GPT-4 to synthesize our own auxiliary instruction tuning data which resulted in better performance, as evidenced by our experimental results. Sample scanpaths for an image-stimulus pair and the corresponding GPT-4 generated response for our prompting strategy are provided in Figure 3.

**Human evaluation of quality of GPT-4 generated transcripts.**    To evaluate the quality of the GPT-4 generated transcripts, we recruited three participants to rate 50 GPT-4 generated transcripts. For each image-stimulus pair corresponding to a GPT-4-generated transcript, we first sampled 5 scanpaths included in the prompt to GPT-4 for generating that transcript (positive samples). Five more scanpaths corresponding to other image-stimulus pairs were also sampled (negative samples).

Hence we obtain 10 scanpaths in total per image-stimulus pair. Each of these 10 scanpaths is overlaid on the given image and presented to three participants to rate (on a scale of 1-5, 1 being very inconsistent and 5 being very consistent) based on the displayed scanpath's consistency with the GPT-4-generated attention allocation description. This assessment showed that the level of hallucination in these GPT-4 generated transcripts is limited, as participants' ratings for positive samples were statistically significantly higher than for negative samples ($p < 0.05$), both in terms of ratings of each individual participant and ratings of all three participants combined. On the scale of 1-5, positive samples were rated on average 3.28 ($\pm 1.29$ ), and negative samples were rated on average 2.04 ($\pm 1.06$). we performed paired samples t-test and regression modeling to account for the between-item response biases. For paired samples t-test, T-statistic = 17.662 and p-value < 0.001, which also suggests that there is a strong and statistically significant difference between scores for positive and negative samples, with positive samples scoring higher on average. Similarly, mixed linear regression modeling of the survey scores revealed that positive samples had scores about 1.25 points higher than negative samples on average (recall that the survey scores were on a scale of 1-5, making this difference quite significant), and this effect is large and statistically significant (z-value = 21.128, $p < 0.001$). Also, group variance (0.092) was small compared to residual variance (1.3024), suggesting that most of the variation is within conditions rather than across items.

## C.3 GPT-4 PROMPT TEMPLATE FOR LLM-AS-A-JUDGE EVALUATION.

We use a comparative prompt comparing multiple methods, as done in Xiong *et al.* (Xiong et al., 2024), by providing the texts generated by them in order to give GPT-4 more context in evaluation. A sample is provided below:

*System Prompt:* You are a decisive evaluation assistant specializing in analyzing images with bounding box annotations and assessing generated expressions. For each case, compare Output A, Output B, Output C, Output D, and Output E against the provided ground truth. Assign distinct scores unless the outputs are truly equivalent. If you judge one output strictly better on any criterion, assign it at least 2 points higher.

*User Prompt*: Given an image described by objects represented as <category: [x, y, w, h]>: <bicycle:[0.153, 0.434, 0.517, 0.49]>, <car:[0.111, 0.25, 0.485, 0.347]>, <car:[0.68, 0.222, 0.236, 0.271]>, <bench:[0.085, 0.617, 0.643, 0.37]>, <car:[0.61, 0.257, 0.157, 0.111]>, <car:[0.428, 0.243, 0.256, 0.155]>, <truck:[0.083, 0.187, 0.199, 0.191]> , please compare the following generated referring expressions: Output A: "the car on the far right", Output B: "right side suv", Output C: "car on right", Output D: "white car to right", Output E: "white car on right" with the ground truth expressions ["benz", "silver benz", "right car"] referring to the object in bounding box [0.68, 0.222, 0.234, 0.269] based on "Expression Overlap" (How much does the generated referring expression match the ground truth referring expressions, especially in terms of coverage of the entities, their attributes and spatial relationships?), "Referential Equivalence" (Does either of the ground truth expressions refer to the same object in the image as the generated expression?) "Category Correctness" (Is the type or category of the referred object mentioned correctly?). Use a scale from 1 to 10 for each criterion, where 1 is poor and 10 is excellent. Format the response in the form of a dictionary with exactly five top-level keys: "Output A", "Output B", "Output C", "Output D" and "Output E". Each of those keys maps to a nested dictionary with exactly the following keys – "Expression Overlap", "Category Correctness", and "Referential Equivalence" – whose values are your numeric scores. Do not include explanations, provide only the scores.

## D IMPLEMENTATION DETAILS

***Gazette.*** We initialize the weights of Gazette from pre-trained LLaVA-1.5-7B (Liu et al., 2024) architecture. This model houses a Vicuna (Chiang et al., 2023) v1.5 LLM and for the vision encoder, and uses a CLIP (Radford et al., 2021) VIT-large image encoder. We train all Gazette variants using LoRA (Hu et al., 2021a) for a maximum of 3 epochs, with a learning rate of 2e-5 (cosine learning rate scheduler), a batch size of 32, and maximum LLM context length of 2048. All code is written in PyTorch (Paszke et al., 2019), and we used DeepSpeed (Rasley et al., 2020) acceleration framework for training. We train our models on two NVIDIA RTX A6000 (48 GB) cores, with each training epoch taking approximately 16 hours to complete. During inference, processing of each scanpath (one inference sample) took approximately 1.2 seconds to complete. For inference-time matching for behavior type $\mathbf{D}_{type}$, and stimulus (target category) for Target-Present and Target-Absent search trials, we use the language encoder, MiniLM (Wang et al., 2020), to encode the generated texts and labels in the label-vocabulary and consequently match the generated texts with labels by computing cosine similarities between language embedding of the generated text and language embedding of each label in the label vocabulary. Finally, we pick the label with the highest cosine similarity. For $\mathbf{D}_{type}$, the label-vocabulary is [“Target-Present Visual Search”, “Target-Absent Visual Search”, “Object Referral”, “Visual Question Answering”]; for target categories in visual search, the label vocabulary is the set of 18 categories in COCO-Search18 (Chen et al., 2021). Note that for VQA samples, the localization idea unit in a think-aloud transcript is simply a placeholder saying “The human is answering a question about the image”, since there are no targets to localize. We use GPT-4o version of GPT-4 to create the think-aloud transcripts and as the evaluation engine under LLM-as-a-Judge evaluation paradigm.

***LLaVA-last.*** *LLaVA-last* is a model that involves predicting the goal based on the last fixation. It is initialized with the weights of Gazette from pre-trained LLaVA-1.5-7B (Liu et al., 2024) architecture. Then following the same hyperparameters as Gazette, we train the model on gaze decoding with this prompt template: `Describe the object at co-ordinate (x,y). Use referring expressions if there are multiple objects of the same category, else just generate the object category which you can choose from [bottle, bowl, car, chair, clock, cup, fork, keyboard, knife, laptop, microwave, mouse, oven, potted plant, sink, stop sign, toilet, tv]`. This template is also used for inference. Note that mentioning the list of categories help for LLaVA-last, but not for Gazette, as revealed by our experiments - perhaps because LLaVA-last is trained on only Object Referral and Target-Present Visual Search, but Gazette is trained on additional Target-Absent Visual Search and VQA, the latter not having any singular target.

**LLaVA-1.5.** This baseline is the frozen, pretrained LLaVA-1.5-7B model (Liu et al., 2024). We provide the same *GazeDec* prompt, with one modification – for visual search scanpaths, we provide the set of COCO-Search18 categories in the prompt, similar to *LLaVA-last*.

**GazeGNN.** We implemented *GazeGNN* (Wang et al., 2024a) for gaze target classification. Nishiyasu *et al.* (Nishiyasu & Sato, 2024) also implemented GazeGNN but this implementation is not public. Specifically, we replace the final classification layer with two classification layers to do the following tasks: (i) predict behavior type (Target-Present or Target-Absent) and (ii) predict stimulus (one of 18 COCO-Search18 categories). This baseline is not extensible to Object Referral and VQA. All hyperparameters are preserved as designated by the authors of GazeGNN.

## E LLM USAGE

This paper employs LLMs in three ways: (1) Our proposed model, Gazette, is based on a pre-trained LLaVA-1.5-7B (Liu et al., 2024) architecture which contains a pre-trained Vicuna (Chiang et al., 2023) v1.5 LLM. (2) We used GPT-4o to derive the think-aloud transcripts for auxiliary instruction tuning of Gazette. (3) We used GPT-4o to refine the grammar of this paper.

# F  QUALITATIVE ANALYSIS

In this section, we qualitatively analyze the model-generated predictions and the attention-allocated explanations generated by Gazette. The qualitative examples are in Fig. 4, Fig. 5, Fig. 6, and Fig. 7.

Fig. 4 shows a scanpath of a human grounding the referring expression "silver benz", and Gazette both trained with or without *ThinkAloud* instructions identify the correct category, *i.e.,* car, but without deeper understanding of the verification and scanning strategies exhibited by the scanpath through *ThinkAloud* instruction tuning, the model misinterprets the "red car" to be the target, whereas our full Gazette model trained on *ThinkAloud* instructions identifies the correct car by analyzing the gaze patterns, as evidenced by the generated attention allocation explanation. The crucial contribution of *ThinkAloud* task is underscored by the model behavior for VQA scenarios, as shown in Fig. 5, where complex reasoning patterns are at play. We see that Gazette trained with *ThinkAloud* interprets the scanpath correctly and manages to reconstruct the question from the scanpath to a great degree, even though it asks about "curtains" instead of the "window" they cover. On the other hand, without *ThinkAloud*, Gazette miserably fails to decode the question corresponding to this scanpath. In Fig. 6, we see a scanpath for a Target-Present search for a "TV". Owing to fixations next to the chair and close to the floor, Gazette trained w/o *ThinkAloud* instructions erroneously predicts a Target-Absent search for a "potted plant". As evidenced by the attention-allocation explanation generated by Gazette trained with *ThinkAloud* instructions, Gazette correctly identifies the target "TV" that is present in the image. Finally, we show a scanpath of Target-Absent for a "bowl" in Fig. 7. We see that Gazette trained with *ThinkAloud* instructions identifies the Target-Absent search for a "bowl" correctly, instead of mispredicting the "fork" distractor object category. We posit that this because of the model finding no fixations to a fork's distractor object in the scene, *i.e.,* spoon, but to the "cup" which is a distractor for a "bowl", as captured within the generated attention allocation explanation.

Gazette Predicts: **Object Referral** for **"black car top right"**.
Gazette w/o ThinkAloud Predicts: Stimulus:**Object Referral** for **"red car"**.

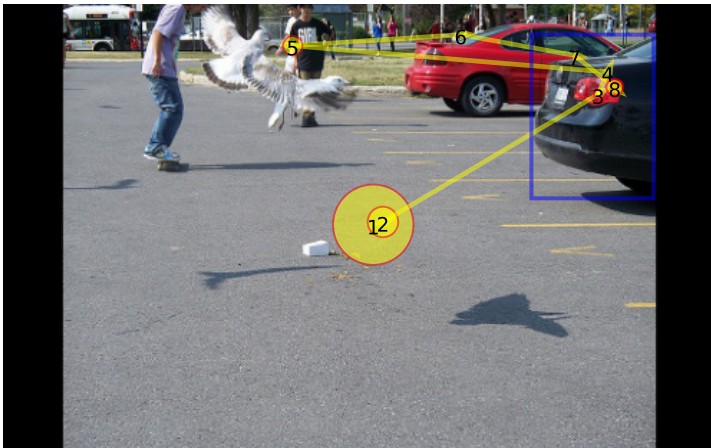

**Gazette-generated attention-allocation explanation**: Most humans initially fixate on the road before shifting their attention to the car located in the upper right corner of the image, often after scanning other cars and occasionally focusing on people in the vicinity.

Figure 4: **Qualitative Results [1/4].** Comparison of methods on decoding a gaze scanpath corresponding to **Object Referral** for referring expression **"black car on right"**. The target object is within the blue bounding box. We provide model predictions from full model Gazette, and its variant not trained on *ThinkAloud* instructions. We also provide the attention allocation explanation idea unit in the Gazette-generated think-aloud transcript.

Gazette Predicts: **VQA** for **"Are there curtains to the right of the chair that is to the right of the lamp?"**
Gazette w/o ThinkAloud Predicts: **VQA** for **"Is there a table to the right of the vase?"**

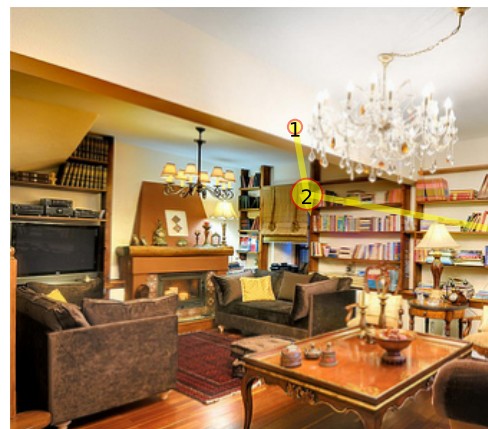

**Gazette-generated attention-allocation explanation**: Most humans fixated on the chandelier and chain, located centrally in the image, before moving to the curtains and window positioned to the right of the chair, with frequent attention to the light and cabinet near the right edge.

Figure 5: **Qualitative Results [2/4].** Comparison of methods on decoding a gaze scanpath corresponding to **Visual Question Answering** for the question **"Is the window behind the chair near the pillows?"**. We provide model predictions from full model Gazette, and its variant not trained on *ThinkAloud* instructions. We also provide the attention allocation explanation idea unit in the Gazette-generated think-aloud transcript.

Gazette Predicts: **Target-Present Search** for **"TV"**.
Gazette w/o ThinkAloud Predicts: **Target-Absent Search** for **"Potted Plant"**.

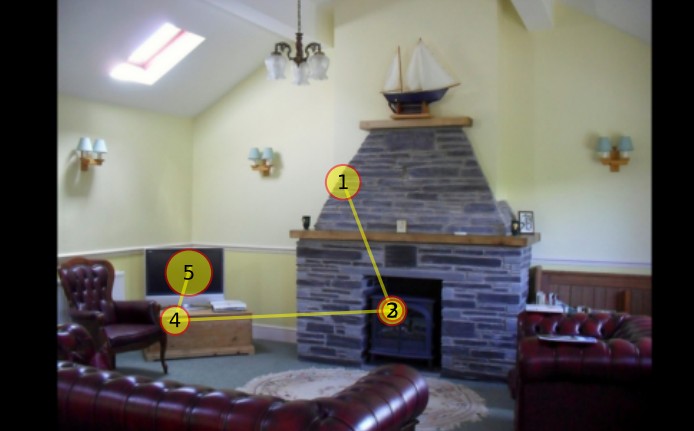

**Gazette-generated attention-allocation explanation**: Most humans initially fixate on the wall or fireplace, then shift their gaze towards the vicinity of the TV, often fixating on or near the TV multiple times, with some eventually locating the TV positioned slightly below and to the right of the center of the image.

Figure 6: **Qualitative Results [3/4].** Comparison of methods on decoding a gaze scanpath corresponding to **Target-Present Search** for a **"TV"**. We provide model predictions from full model Gazette, and its variant not trained on *ThinkAloud* instructions. We also provide the attention allocation explanation idea unit in the Gazette-generated think-aloud transcript.

Gazette Predicts: **Target-Absent Search** for **"Bowl"**
Gazette w/o ThinkAloud Predicts: **Target-Absent Search** for **"Fork"**.

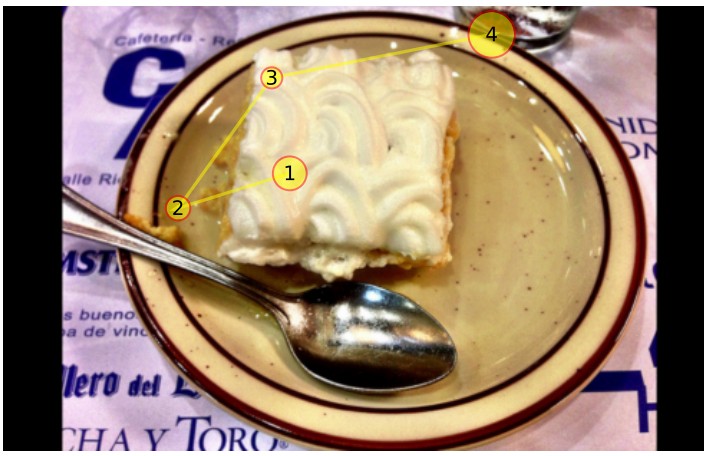

**Gazette-generated attention-allocation explanation**:Most humans initially fixate on the large cake in the center of the image before shifting their gaze to the cups located on the upper right side, often returning to the cake or intermittently fixating on the dining table.

Figure 7: **Qualitative Results [4/4].** Comparison of methods on decoding a gaze scanpath corresponding to **Target-Absent Search** for a **"bowl"**. We provide model predictions from full model Gazette, and its variant not trained on *ThinkAloud* instructions. We also provide the attention allocation explanation idea unit in the Gazette-generated think-aloud transcript.

