# OpenReview forum: "Gaze-to-text Generation: Beyond Categorical Decoding of Human Attention"
_ICLR.cc/2026/Conference — ICLR 2026 Conference Withdrawn Submission_

### Official Review · Reviewer_zZvY · 2025-10-31

**Soundness:** 1
**Presentation:** 2
**Contribution:** 2
**Rating:** 2
**Confidence:** 4

**Summary:**

This paper proposes Gazette, which takes an image along with a sequence of coordinates (representing the gaze scanpath within the image) as input, and outputs a natural language description of that gaze scanpath. The authors use GPT-4 to generate natural language descriptions as ground truth to train the LLaVA-1.5-7B model, and evaluate the system by measuring the similarity between the model-generated descriptions and the GPT-4 generated ground truth (with metrics like BLEU and LLM-as-a-Judge).

**Strengths:**

- The idea of converting gaze into text is intuitively reasonable and meaningful.

**Weaknesses:**

- The experimental objective and design are incorrect. Since the paper claims to be the first to propose a gaze-to-text framework, the focus of the experiments should be on validating the advantages of representing gaze as text, rather than merely evaluating the accuracy of the generated descriptions. Under this premise, the authors should design downstream tasks that require gaze understanding and compare the performance of Gazette with other gaze-analysis algorithms. A VQA dataset that incorporates gaze would be a more appropriate benchmark than textual similarity.  A very straightforward baseline (which is not sufficient but serves as a direct example) would be to use VLMs or object detection algorithms to identify objects at the key points of the gaze scanpath and use the resulting object sequence as an additional input to assist VLMs in performing VQA. In addition, the quality of the GPT-4 generated ground truth is not verified, which undermines the reliability of both the model’s performance and the experimental evaluation.
- The method design is simplistic. Gazette uses a large, closed-source model to generate data and then trains a smaller, open-source model. This is very common and, to some extent, even outdated. The only additional design introduced by Gazette is splitting the output text into coarse-grained and fine-grained descriptions, which is also not a novel idea.
- The paper lacks necessary analytical experiments, especially regarding how gaze scanpath are fed into VLMs. Although many VLMs can handle coordinates within an image, this capability is not effectively validated. The authors should compare the current input method with alternative approaches, including (but not limited to) directly annotating the gaze scanpath on the image.

**Questions:**

See the weaknesses section. If the first and third issues can be addressed, I would be willing to raise my score.

---

> ### Author Response · Authors · 2025-11-13
> **Authors' response to Reviewer zZvY**
>
> 1. **Response to "The experimental objective and design are incorrect."**:
> The reviewer’s interpretation reflects a misunderstanding of the name of our method. The term Gaze-to-text refers to our output format—the model decodes gaze into text—not to the way gaze is provided to the model. Our task is to model gaze movements within an MLLM framework to infer the goal that drives those movements, which is a non-trivial reasoning problem that goes far beyond merely representing gaze as text. While extending our model to downstream tasks that require gaze understanding, as suggested by the reviewer, is a promising direction, it is out of scope for our paper which focuses on this non-trivial unconstrained gaze decoding task.
>
> 2. **Response to “Paper lacks necessary analytical experiments, especially regarding how gaze scanpaths are fed into VLMs. Although many VLMs can handle coordinates within an image, this capability is not effectively validated.”**:
> The reviewer’s claim that VLMs’ ability to handle coordinates is “not validated” is contradicted by prior work. Ranasinghe et al. [1] provide an extensive analysis of how VLMs process image coordinates and localization. Moreover, Ranasinghe et al. [1], Liu et al. [2], and Li et al. [3] show that representing 2D coordinates and scene objects textually is both effective and efficient in MLLMs. Following this established precedent (as we have mentioned in Lines 171–173), we represent gaze coordinates and fixation durations as text.
>
>    Directly annotating scanpaths on the image is suboptimal for two reasons:
>     (1) multiple fixations may often coincide at the same location within the 336×336 resolution used in LLaVA-1.5, especially for AiR-D and RefCOCO-Gaze, making visual encoding unreliable; and
>     (2) fixation duration and temporal order, which are crucial aspects of scanpaths, would be lost in these image-level annotations.
>     Our textual encoding thus follows prior research and avoids the limitations of image-only representations.
>
> 3. **Response to “The quality of the GPT-4 generated ground truth is not verified”**:
> We already provide a human evaluation of the GPT-4–generated ground truth. This evaluation is documented in the Appendix (Lines 1022–1041) and referenced in the main text (Lines 278 and 463). The reviewer’s concern appears to overlook this analysis.
>
> 4. **Response to “Method design is simplistic”**:
> Our technical contribution is not an architectural overhaul—which is neither practical nor appropriate in data-scarce domains such as human gaze behavior. Instead, our contributions are: (1) extending gaze decoding to open-ended generative goals via the Gazette framework, and (2) introducing think-aloud transcript generation, leveraging GPT-4–generated attention-allocation patterns to enhance top-down attentional reasoning.
>
>     Large pretrained LLM/MLLM backbones are extremely costly to retrain and susceptible to catastrophic forgetting. Given the limited amount of gaze data available, architecture-level modifications are not a viable or reliable strategy. Our approach is therefore intentionally data-centric: we synthesize high-quality instruction-style examples using a novel prompting strategy that differs from prior synthetic-data pipelines, and fine-tune a pretrained MLLM backbone while preserving its existing competencies. Our empirical results show significant improvements owing to this contribution.
>
>     This design choice is strongly supported by LLaRA (Li et al. [3], ICLR 2025), which achieves state-of-the-art performance by retaining the pretrained VLM (LLaVA) and generating synthetic visuo-textual conversation-style visuomotor policies rather than altering the architecture. LLaRA outperforms methods such as RT-2 [4] that rely on architectural modifications (e.g., action tokenization), particularly under limited robot-data regimes. Our work follows the same principled rationale, adapted to the domain of human gaze decoding.
>
> References:
>
> [1] Kanchana Ranasinghe, Satya Narayan Shukla, Omid Poursaeed, Michael S Ryoo, and Tsung-Yu Lin. Learning to localize objects improves spatial reasoning in visual-llms. In Proceedings of the IEEE/CVF Conference on Computer Vision and Pattern Recognition, pp. 12977–12987, 2024.
>
> [2] Haotian Liu, Chunyuan Li, Qingyang Wu, and Yong Jae Lee. Visual instruction tuning. Advances in neural information processing systems, 36:34892–34916, 2023.
>
> [3] Xiang Li, Cristina Mata, Jongwoo Park, Kumara Kahatapitiya, Yoo Sung Jang, Jinghuan Shang, Kanchana Ranasinghe, Ryan Burgert, Mu Cai, Yong Jae Lee, et al. Llara: Supercharging robot learning data for vision-language policy, ICLR 2025
>
> [4] Zitkovich, Brianna, Tianhe Yu, Sichun Xu, Peng Xu, Ted Xiao, Fei Xia, Jialin Wu et al. "Rt-2: Vision-language-action models transfer web knowledge to robotic control." In Conference on Robot Learning, pp. 2165-2183. PMLR, 2023.

---

### Official Review · Reviewer_wNDR · 2025-11-01

**Soundness:** 2
**Presentation:** 2
**Contribution:** 2
**Rating:** 4
**Confidence:** 1

**Summary:**

This work introduces decoding gaze into natural language descriptions of human goals across diverse visual tasks. Unlike prior work, which frames gaze decoding as a classification task over predefined categories, this paper formulates it as a generative learning problem: training a model to produce free-form descriptions that capture the rich context, nuance, and open-ended nature of human intentions beyond fixed labels.

**Strengths:**

1. This paper introduces the task of unconstrained decoding of goal-directed attention, where top-down goals are expressed in natural language, supporting a wide range of human gaze behaviors.
2. This paper proposes a text-generative MLLM-based framework, an instruction-tuned to decode a scanpath of gaze fixations (during image viewing) to natural language.

**Weaknesses:**

1. The motivation of this work is not very clear. For example, why is decoding gaze as natural language better than simple labels?
2. The innovation of the proposed model is quite limited. For example, the explanation of gaze seems to derive from the prompting strategy of LLMs.
3. The data used to generate natural language for gaze is not clear.

**Questions:**

1. How is the data for natural language decoding obtained?
2. Why is decoding gaze as natural language better than simple labels?
3. The use of gaze information in the model is based on prompting?
4. What's the relationship between gaze and visual representation?

---

> ### Author Response · Authors · 2025-11-13
> **Authors' response to Reviewer wNDR**
>
> 1. **Response to Weakness 1 and Question 2**:
> We have discussed in Lines 41–51 why decoding gaze into natural language offers advantages over using simple categorical labels. We will further clarify that although categorical labels may suffice in certain idealized settings, they often fail to capture the nuance required in real-world scenarios. Our example in Lines 41–51—analyzing user engagement on an e-commerce website—illustrates this point. In practice, free-form language descriptions support far more precise awareness modeling than coarse labels such as “car” or “chair.” Natural language provides richer contextual information, enabling the model to infer detailed intent rather than relying on oversimplified category labels.
>
> 2. **Response to Weakness 2 and Question 3**:
> We emphasize that our model generates the explanations itself. GPT-4 is used only to produce the ground-truth supervision needed to train the model on the auxiliary talk-aloud transcript generation task. The reviewer’s concern appears to misunderstand the model’s outputs versus the training signal; we clarify that transcripts can be solely generated by our model at inference time.
>
> 3. **Response to Weakness 3 and Question 1**:
> The natural-language goals associated with gaze scanpaths are derived directly from existing gaze datasets—COCO-Search18, RefCOCO-Gaze, and AiR-D—as described in Section 4 (Lines 303–308).
>
> 4. **Clarification for Question 4**:
> Across COCO-Search18, RefCOCO-Gaze, and AiR-D, gaze data captures how humans allocate visual attention during both search and language-guided tasks. This provides a direct link between human eye movements and visual semantic representations. Leveraging gaze in this way enables our model to better align perception, attention, and high-level semantics, which is central to understanding human-driven visual reasoning.

---

### Official Review · Reviewer_hS2s · 2025-11-03

**Soundness:** 2
**Presentation:** 3
**Contribution:** 2
**Rating:** 2
**Confidence:** 4

**Summary:**

The paper proposes Gaze-to-text Generation, shifting from classifying gaze to generating natural-language descriptions of a user’s intention from their scanpath and the viewed image. The authors introduce Gazette, an MLLM-based framework (built on LLaVA-1.5-7B) that textualizes gaze and outputs a “cognitive context” describing behavior type (search, referral, VQA) and the specific goal or stimulus. They also synthesize think-aloud supervision with GPT-4 from multiple observers’ scanpaths on the same task, teaching the model to focus on task-driven spatiotemporal patterns.

**Strengths:**

1. Propose a new task Gaze-to-text Generation
2. Generate a dataset of Gaze-to-text Generation using ChatGPT
3. Finetune a LLaVA1.5 model to perform the task.

**Weaknesses:**

1. The necessity of the proposed new task Gaze-to-text Generation is quetionable. If the gaze points are already there, an additional referring which generates the text will lead to a structure gaze movement. Is natural language really necessary to describe the gaze movement? I think structured representation contains enough information to infer intention or study human behavior.
2. Do we really need to finetune a MLLM to perform this tasks or we can achieve the same performance with assembled set of foundations models off-the-shelf. One example, is using referring models to extract what each gaze point is, then directly summarize with a LLM. This could serve as a strong baseline which is missing in the paper.
3. Lack of technical contribution. This paper only proposes a task with a generated dataset and then finetunes a LLaVA without making any adaptions to LLaVA for the task. All these efforts are largely reuses existing techniques (e.g. using ChatGPT to generate data is proposed in the LLaVA paper and has been widely adopted, LLaVA model is also widely used and there is no technical improvement in this paper to improve LLaVA).

**Questions:**

Please see Weaknesses.

---

> ### Author Response · Authors · 2025-11-13
> **Authors' Response to Reviewer hS2s**
>
> 1. **Response to Weakness 1**:
> The reviewer’s concern reflects a misunderstanding of our task. We are not attempting to describe the gaze scanpath itself; we aim to **infer/decode the goal** that produced the gaze scanpath. For example, given the question “Are there curtains to the right of the chair that is to the right of the lamp?” (Figure 5, appendix), the scene may contain no curtains at all, and thus no gaze fixation can correspond to that object. Nevertheless, the model must still decode the latent goal—the underlying question being pursued. A VLM that merely describes the gaze points will necessarily fail in such cases, including in target-absent search, where no gaze point falls on the goal target as it is is always absent in the scene. Our method is explicitly designed to recover this latent intention, not to simply paraphrase the objects traversed by gaze trajectories.
>
> 2. **Response to Weakness 2**:
> We already include two relevant baselines: *LLaVA-1.5* and *LLaVA-last*. For Object Referral and Target-Present Visual Search, last fixations typically fall on the target; therefore, LLaVA-1.5 is fine-tuned to describe the object at the final fixation, yielding the **LLaVA-last** baseline. We also evaluate a frozen **LLaVA-1.5** (Liu et al., 2024) prompted with the scanpath and instructions to describe the goal. Importantly, both baselines perform substantially worse than our method, Gazette, across all decoding metrics (Tables 1–3). This demonstrates that naïvely describing fixations or prompting a strong VLM is insufficient for the task. We would like to clarify again that we are not attempting to describe the gaze scanpath itself; we aim to **infer/decode the goal** that produced the gaze scanpath. Hence using referring models to extract what each gaze point is, then directly summarizing with a LLM, as the reviewer suggested, is not a suitable solution for our task.
>
> 3. **Response to Weakness 3**:
> Our technical contribution is not an architectural overhaul—which is neither practical nor appropriate in data-scarce domains such as human gaze behavior. Instead, our contributions are:
> (1) extending gaze decoding to open-ended generative goals via the Gazette framework, and
> (2) introducing think-aloud transcript generation, leveraging GPT-4–generated attention-allocation patterns to enhance top-down attentional reasoning.
>
>     Large pretrained LLM/MLLM backbones are extremely costly to retrain and susceptible to catastrophic forgetting. Given the limited amount of gaze data available, architecture-level modifications are not a viable or reliable strategy. Our approach is therefore intentionally data-centric: we synthesize high-quality instruction-style examples using a novel prompting strategy that differs from prior synthetic-data pipelines, and fine-tune a pretrained MLLM backbone while preserving its existing competencies. Our empirical results show significant improvements owing to this contribution.
>
>     This design choice is strongly supported by LLaRA (Li et al. [1], ICLR 2025), which achieves state-of-the-art performance by retaining the pretrained VLM (LLaVA) and generating synthetic visuo-textual conversation-style visuomotor policies rather than altering the architecture. LLaRA outperforms methods such as RT-2 [2] that rely on architectural modifications (e.g., action tokenization), particularly under limited robot-data regimes. Our work follows the same principled rationale, adapted to the domain of human gaze decoding.
>
> References:
>
> [1]  Xiang Li, Cristina Mata, Jongwoo Park, Kumara Kahatapitiya, Yoo Sung Jang, Jinghuan Shang, Kanchana Ranasinghe, Ryan Burgert, Mu Cai, Yong Jae Lee, et al. Llara: Supercharging robot learning data for vision-language policy, ICLR 2025
>
> [2] Zitkovich, Brianna, Tianhe Yu, Sichun Xu, Peng Xu, Ted Xiao, Fei Xia, Jialin Wu et al. "Rt-2: Vision-language-action models transfer web knowledge to robotic control." In Conference on Robot Learning, pp. 2165-2183. PMLR, 2023.

---

### Note · Authors · 2025-11-14

I have read and agree with the venue's withdrawal policy on behalf of myself and my co-authors.